# HIV Latency in Myeloid Cells: Challenges for a Cure

**DOI:** 10.3390/pathogens11060611

**Published:** 2022-05-24

**Authors:** Alisha Chitrakar, Marta Sanz, Sanjay B. Maggirwar, Natalia Soriano-Sarabia

**Affiliations:** Department of Microbiology, Immunology and Tropical Medicine, The George Washington University, Washington, DC 20037, USA; alishachitrakar@gwu.edu (A.C.); martasanzp@gwu.edu (M.S.); smaggirwar@gwu.edu (S.B.M.)

**Keywords:** HIV latency, HIV cure, cellular reservoirs, monocytes, macrophages, CNS, myeloid cells

## Abstract

The use of antiretroviral therapy (ART) for Human Immunodeficiency Virus (HIV) treatment has been highly successful in controlling plasma viremia to undetectable levels. However, a complete cure for HIV is hindered by the presence of replication-competent HIV, integrated in the host genome, that can persist long term in a resting state called viral latency. Resting memory CD4+ T cells are considered the biggest reservoir of persistent HIV infection and are often studied exclusively as the main target for an HIV cure. However, other cell types, such as circulating monocytes and tissue-resident macrophages, can harbor integrated, replication-competent HIV. To develop a cure for HIV, focus is needed not only on the T cell compartment, but also on these myeloid reservoirs of persistent HIV infection. In this review, we summarize their importance when designing HIV cure strategies and challenges associated to their identification and specific targeting by the “shock and kill” approach.

## 1. Introduction

### Definition of Viral Reservoir

Prevention of HIV replication by simultaneously targeting several steps in the viral cycle with antiretroviral therapy (ART) allows to reduce plasma viremia below the limit of detection in clinical assays. However, ART does not eliminate HIV, and after depletion of CD4+ T cells during primary infection, some cells enter a resting state where HIV can remain persistently integrated into the host cell genome [1,2,3]. This resting state, where HIV is not actively replicating, is called viral latency. Therefore, viral reservoirs are defined as cells that harbor integrated replication-competent HIV that persist in long-lived cells during suppressive ART. Multiple mechanisms are involved in establishing such latency and have been discussed elsewhere [4]. Resting memory CD4+ T cells are the most widely studied cellular reservoir of persistent HIV and they are extremely stable over time [5,6]. HIV persists in all subsets of CD4+ T cells including memory and naïve cell subpopulations although distinct transcriptional activity and integration sites can contribute to variable viral inducibility following reactivation [7,8,9,10,11]. Contributors to HIV persistence within CD4+ T cells during ART include homeostatic or antigen-driven proliferation and possibly ongoing viral replication that maintains the immune system activated and exhausted [7,12,13,14,15,16,17,18,19]. These elements constitute the main barrier to cure.

In addition to CD4+ T cells, other cell types might serve as HIV reservoirs, including γδ T cells that constitutively express CCR5 [20]. Despite generally lacking CD4 expression, γδ T cells transiently upregulate CD4 expression upon activation, become targets for HIV infection and can serve as reservoirs harboring replication-competent HIV [21,22]. In addition, other non-T cell subpopulations, including monocytes and macrophages, are susceptible to HIV infection and allow long-term persistence of HIV in people living with HIV (PLWH) on optimal ART. This article reviews main characteristics of human circulating and tissue-resident myeloid cells, their role as reservoirs and potential issues for current strategies to cure HIV infection. Finally, we discuss challenges and strategies that may help overcome killing resistance by effector cytotoxic cells as part of the shock and kill approach.

## 2. Myeloid Cells: Origin, General Characteristics and Subpopulations

Hematopoietic stem cells in the bone marrow produce two different progenitor cells that will give raise to the lymphoid and myeloid lineages (Figure 1). Immature myeloid cells are intermediate precursors of the normal process of myelopoiesis that comprise a heterogeneous population of common myeloid progenitor cells that rapidly differentiate into mature myeloid cells. However, in several pathological conditions, immature myeloid cells with suppressive activity, known as myeloid-derived suppressor cells, are expanded and mediate suppression of T cell functions [23,24].

Mature myeloid cells belong to the innate immune system and comprise several subpopulations classified into mononuclear and polymorphonuclear cells (or granulocytes). Mononuclear phagocytes include monocytes, macrophages, and dendritic cells (DCs), and granulocytes include neutrophils, eosinophils, mast cells and basophils [25]. Granulocytes, and neutrophils specifically, are the most abundant leukocytes with critical antimicrobial functions. As the first responders to infection, they are rapidly recruited to sites of infection and they are short-lived with a circulating half-life of 6–8 h [26]. Monocytes constitute the circulating precursors of macrophages, which reside virtually in all tissues. Monocytes play a major role in protective immunity by circulating through the blood and the lymphatic system, where they are recruited to sites of tissue injury, activated to secrete inflammatory cytokines and to phagocyte [27]. Tissue-resident macrophages perform immune surveillance, homeostatic and tissue repair functions [28,29]. Monocytes had been long considered a developmental intermediate between precursors of the bone marrow and mature tissue macrophages. However, it has become clear now that both circulating monocytes recruited to tissues contribute to in situ homeostasis [30,31], and many tissue-resident macrophage populations locate in those tissues and are homeostatically maintained [31,32,33,34,35]. Early observations have shown that macrophages exist in the embryonic yolk sac prior to the formation of hematopoietic stem cells and therefore are seeded within tissues during embryogenesis [36]. However, that origin has been neglected for decades and the idea that blood-derived monocytes were the only cells responsible for tissue-macrophage replenishment prevailed leading to misconceptions on monocyte-macrophage origins and functions. Currently, the embryonic origin of macrophages is widely demonstrated and accepted [33,37,38,39]. In addition, embryonic cells with myeloid cell potential can migrate to the fetal liver and differentiate into progenitor cells and are considered the key progenitor cells for developing tissue-specific macrophages [36,37].

This improved understanding of the myeloid population immunobiology and function has been fundamental to expand investigations towards characterizing how the HIV reservoir is established, maintained and reactivated in specific monocytes and macrophage populations. The relevance of such a reservoir remains poorly characterized and understood despite being a critical component for HIV cure strategies due to their longevity and self-renewal properties.

## 3. Circulating Monocytes and Tissue-Resident Macrophages Subpopulations and Characteristics

Circulating monocytes can be identified by flow cytometry using CD14 and CD16 monoclonal antibodies. They are primarily classified into three phenotypically and functionally distinct subpopulations: classical (90%, CD14^++^CD16^−^), intermediate (5%, CD14^++^CD16^+^) and non-classical (5%, CD14^+^CD16^++^) monocytes [40,41,42]. Further work has defined additional monocyte subpopulations with specific epigenetic and cytokine production signatures, providing an extra layer of complexity within the monocyte lineage [43,44]. In addition, age and sex may impact the interindividual distribution of monocyte/macrophage subpopulations with specific phenotype and functions [43,44,45,46], and was recently reviewed [47,48]. Immune cells’ trafficking to be recruited to sites of injury, infection or transformation is governed by the level of chemokine receptors’ expression on a given cell population, and the balance of chemokine and cytokine levels present in the environment [49,50,51,52]. As such, phenotypic signatures of the three primary monocyte subpopulations govern their migration capabilities based on expression levels of chemokine receptors.

Classical monocytes express high levels of CCR2 and low levels of CX3CR1 and migrate in response to CCL2 chemokine gradients, while CD16^+^ monocytes express low CCR2 and high CXCR1 levels and migrate towards CXCL1 (fractalkine) gradients [53,54]. Intermediate monocytes have lower expression of CCR2 and CXCR1, and higher expression of CX3CR1 compared to classical monocytes. MHC-II expression is higher in intermediate monocytes than both the classical and non-classical population. Expression patterns of these chemokine receptors among the three main monocyte subpopulations are linked to specific functions. Classical monocytes migrate to sites of injury and differentiate into inflammatory macrophages, whereas non-classical monocytes exhibit immune surveillance functions along the blood vessels. Both non-classical and intermediate monocytes are the source of proinflammatory cytokines, such as TNF-α, IL-1β, IL-6 and IL-18, with intermediate monocytes producing significantly more TNF-α and IL-1β than the other populations.

Tissue-resident macrophages have critical roles as first line of defense against intruding pathogens, maintaining homeostasis, and tissue integrity. Their phagocytic function is crucial for both uptake and clearance of injured, dying or cellular debris, and pathogenic microbes [28,55]. Located in different anatomical areas and organs, macrophages require tight regulation by transcriptional mechanisms, regulated both by origin and tissue microenvironment [56]. Some tissue-resident macrophage populations, including microglia and Langerhans’ macrophages, have self-renewal capacities and therefore can be maintained independently of monocytes, although circulating monocytes migrate to tissues and can acquire the phenotype of resident macrophages [28,34,57,58,59,60]. The extent of the contribution of tissue-resident vs. circulating monocytes to the maintenance of local macrophage populations is currently unknown since trauma, injury or chronic inflammation can affect monocyte infiltration. It seems clear though that the half-lives of certain macrophage populations can be similar to memory T cells and therefore their contribution to the persistence of HIV during suppressive ART and their critical importance for HIV cure strategies must be addressed.

## 4. HIV Infection and Reservoirs in Monocytes/Macrophages

Although CD4+ T cells are the main target of HIV infection, macrophages express low levels of CD4 and their susceptibility to productive infection was described more than 2 decades ago [61]. However, due to the diversity of macrophage subpopulations, relatively low virus production and insufficient understanding of their development, immunobiology and diversity has dampened our general understanding of the role of monocytes/macrophages in HIV pathogenesis [48,62]. Macrophages likely play two opposing roles in the acute phase of HIV infection. On the one hand, they help to establish infection at sites of viral entry, and monocytes and perivascular macrophages disseminate the virus throughout the body, including the brain [63]. The description of tissue-resident macrophages, wider understanding of their self-renewal properties and increased resistance to cytopathic effects compared to CD4+ T cells, has renewed the interest in the field to investigate their role as latent HIV reservoirs. Since permissiveness to HIV infection differs among macrophage populations, depending on the localization and sample size, the identification of latent macrophage reservoirs might be challenging [64,65,66,67].

## 5. HIV Latency in Monocytes

Circulating monocytes in the blood have a shorter lifespan (~4 to 7 days) and they differentiate into macrophage phenotype soon after their extravasation into the tissues [68]. Their role in maintaining a long-term persistent HIV reservoir has therefore been controversial [69]. However, it is well established that monocytes are infected in the acute stages of HIV infection and they play a role in establishing HIV-associated neurocognitive disorder (HAND) [70]. Classical monocytes, expressing low levels of CD4 and CCR5 receptors, are less susceptible to HIV infection than intermediate or non-classical monocytes. Both monocyte subpopulations expressing CD16, have higher levels of CCR5 expression, which makes them more susceptible to HIV infection [71,72]. Therefore, identification of the HIV reservoir within monocytes requires a deeper phenotypic characterization.

Early and late, but not integrated, HIV reverse transcripts were detected in blood monocytes of a small population of ART- suppressed PLWH, whereas HIV DNA was only found in CD4+ T cells [69]. Monocytes have host-restriction mechanisms that limit HIV infection, including SAMHD1 and APOBEC3 [73]. These restriction factors are proteins that provide an initial line of defense against HIV infection [74,75]. Despite the presence of these restriction factors in myeloid cells, integrated HIV DNA and replication-competent virus were detected in some studies performed in PLWH on suppressive ART [76,77,78].

A recent study investigating cellular reservoirs of HIV-1 in donors undergoing analytic treatment interruption (ATI) also identified macrophage-tropic HIV-1 variants in the plasma of ART-suppressed PLWH. Molecular clock analysis also suggested that these variants were potentially established before ART interruption [79]. These studies present us with further evidence that macrophages need to be considered as critical cellular reservoirs in HIV cure strategies.

## 6. HIV Latency in Tissue-Resident Macrophages

Even though the role of circulating monocytes as a reservoir of HIV is debatable, tissue-resident macrophages have long known to be cellular reservoirs because of their presence in multiple tissues and their long lifespan.

Gut macrophages were previously believed to be resistant to HIV infection and were thought to play a role in generating antibody responses against the virus to decrease plasma viremia in the initial stages of infection [80]. However, using gut biopsy samples from ART-suppressed PLWH, studies have shown that duodenal macrophages express p24 and harbor proviral DNA [38,81]. Similarly, urethral macrophages in ART-suppressed PLWH were also shown to be the primary cells to harbor HIV-1 reservoir in penile urethra [82]. In these macrophages, containing total and integrated HIV-1 DNA, LPS stimulation reversed latency leading to expression of HIV-1 p24 protein [82,83].

HIV-1 DNA was also detected in specialized macrophage Kupffer cells from the liver of PLWH postmortem, although studies in SIV-infected macaques suggest that the liver is not the primary site of viral replication [84,85]. Using bronchoalveolar lavage (BAL) of ART-suppressed PLWH, HIV RNA and proviral DNA was detected in alveolar macrophages in some but not other patients [86,87]. The presence of proviral DNA in alveolar macrophages displayed impaired phagocytic capacity compared to healthy controls, which is consistent with the observations in ART naïve PLWH [86,88]. This suggests the latent alveolar macrophages are impairing pulmonary immunity, making PLWH more susceptible to respiratory tract infections [86,89].

There have been contradicting reports on the role of Langerhans cells during HIV infection. Numerous reports have shown that Langerhans cells can uptake HIV and were previously hypothesized to mediate viral transmission to CD4+ T cells [82,90,91,92]. However, recent studies have contradicted these reports by showing Langerhans cells prevented HIV infection via actions of C-type lectin Langerin, thus preventing further transmission [93]. Regarding the role of Langerhans as additional HIV reservoirs, studies concluded that they may not be a principal reservoir, although some evidence suggests that this may be HIV-subtype dependent [94,95]. Vaginal epithelial dendritic cells (CD1a^+^ VEDCs) were recently defined to be a separate subset that were previously misclassified as Langerhans cells. CD1a^+^ VEDCs isolated from virologically suppressed women harbored HIV-1 DNA, suggesting their potential as an HIV reservoir [96]. The most significant monocytic reservoirs of HIV were noted in the CNS, which will be discussed in detail below.

Collectively, there is significant evidence for the role of macrophages as reservoirs in ART-suppressed individuals that warrants further studies to characterize them in ATI interventions aimed at HIV eradication.

## 7. Models to Studying Reactivation of Myeloid HIV Reservoirs

Investigation of HIV latency in human tissue-resident macrophages is challenging due to their anatomical location that is limited to biopsies post-surgery or postmortem samples. Despite the inherent differences with primary cells, in vitro studies from latently infected monocytic cell lines and animal models are useful as a proxy to better understand mechanisms of viral latency and reactivation.

### 7.1. Latently Infected Monocytic Cell Lines and Primary Cells

Infection of some myeloid cell lines leads to survival of those carrying integrated proviruses [97]. Here, we briefly review some of the latently infected cell lines and reports on susceptibility to reactivation.

U1 promonocytic cells, developed by Fauci’s group, are derived from the U937 parent cells infected with replication-competent HIV-1 (LAV-1 strain) that contain two copies of integrated proviral DNA per cell. These U1 cells have minimal constitutive expression of HIV but virus can be produced upon stimulation with phytohemagglutinin (PMA) and proinflammatory cytokines such as interleukin (IL) IL-1α, IL-1β, IL-6 and TNFα [98,99,100].

The monocytic cell line (THP-1) was derived by culturing the blood of a patient with acute monocytic leukemia by Tada’s group [101]. Upon stimulation with PMA, THP-1 cells have a monocyte-derived macrophage (MDM)-like phenotype with an activation of CD16 and IL-1β. Constitutive expression of CD4, CCR5 and CXCR4 makes them susceptible to HIV-1 infection [97,102,103]. This cell line has been used to describe viral latency and to test reactivation of the virus using LPS, PMA, TNFα and GM-CSF. Transcriptional silencing can be reverted inhibiting DNA methylation by 3-deaza-adenosine [97,104,105].

HC69 are latently HIV-infected human microglial cell lines developed by Karn’s group. They were developed by immortalizing human microglial cells (C20 cell line) using simian virus 40 large T antigen (SV40Tag)/human telomerase reverse transcriptase (hTERT) [106]. These immortalized C20 cells were then infected with a vesicular stomatitis virus G (VSV-G) envelope pseudotyped lentivirus vector (PHR1′/d2EGFP), thus expressing a green fluorescent protein (d2EGFP) as a reporter [106]. Although these cells exhibit a background level of spontaneous GFP expression in culture, latency of HC69 cells can be effectively reversed by using TNFα and poly (I:C) TLR3 agonist [106,107].

Human primary monocyte models may constitute more relevant in vitro models for evaluation of HIV cure strategies [108]. Monocytes isolated from peripheral blood mononuclear cells (PBMC) can be differentiated in vitro into macrophages (MDM) using macrophage colony stimulating factor (MCSF) or granulocyte macrophage colony stimulating factor (GM-CSF) to induce anti- or pro-inflammatory macrophage phenotypes [108,109,110,111]. MDMs can also use other stimuli to polarize them, such as IFNγ and TNFα for M1 (classical) or IL-4 for M2 (non-classical) phenotype [108,112,113,114,115]. Different polarization stimuli can influence HIV infectivity and reactivation in these models [108,112,113,114,115].

The polarized MDMs can be infected with M-tropic eGFP tagged reporter HIV to generate latency models in vitro [108,116]. Brown et al. described an MDM model where these infected MDMs were cultured for up to 78 days post infection and observed that the FACS purified GFP- MDMs harbored proviral DNA, which could be reactivated by IL-4 and PMA to produce replication-competent HIV [116]. Wong et al. recently developed an improved primary model where they generated latently infected MDM model by culturing the infected MDMs with T20 (enfuvirtide) to prevent de novo infection and FACS-sorted GFP-MDMs nine days after infection did not show HIV-1 p24 production. These sorted GFP-MDMs could, however have a background level of spontaneous reactivation in culture. They also showed that the polarizing stimulus before infection modulates the latency reactivation, with M2 polarized MDMs reactivating at a higher rate than M1 polarized MDMs [108]. The latently infected MDMs were used to study modulation of latency reversing agents and are discussed in a section below [108].

These cell lines and in vitro models of latent infection have been popular due to their ease to culture, short time commitment and cost effectiveness. They have been important in demonstrating effectiveness and mechanisms of various molecules used as ART and LRA. However, they are not sufficient for studying systemic effects of targeted therapies, which would require the use of animal models. Especially for studying tissue-resident macrophage reservoirs of HIV, the use of animal models is irreplaceable.

### 7.2. Humanized Mice

Humanized mice are generated using immunodeficient mice transplanted with human cells, tissues or both, therefore, allowing to be HIV infected while recapitulating some aspects of the human immune system. These models allow experimental interventions and tissue sampling that are not possible in clinical studies. This is particularly true for the bone marrow–liver–thymus (BLT) mouse where both CD4+ T cells and myeloid cells can be investigated in the context of HIV infection, pathogenesis and cure. One critical limitation of studies involving humanized mice models is the small size of the mice requiring, in general, pooling of several animals to achieve enough sensitivity to quantify viral reservoirs. The different types of humanized mouse models used to study HIV persistence and their benefits and limitations have been recently reviewed and are only briefly described here [117,118,119].

Peripheral blood lymphocyte (PBL)-humanized (hu)-mice can be generated by reconstituting various strains of immunocompromised mice, such as NOD/SCID/IL2Rγ-null (NSG), lacking mature T, B or NK cells and transplanting with human PBMCs. These PBL-hu mice show reconstitution of CD4+ and CD8+ lymphocytes within 6–8 weeks of transplant [120]. PBL-hu-mice are an effective model because of the ease and time of generation. They provide a good model for HIV infection and prevention, but since they are susceptible to rapid Graft-versus-Host disease (GvHD), they cannot be used as a long-term infection model [121]. Human macrophages are also not incorporated in this model, which makes them a poor model for studying macrophage HIV reservoirs [120,122].

Various strains of immunocompromised mice can be reconstituted with human CD34+ hematopoietic stem cells (HSC) from multiple sites of origin (fetal liver and thymus tissues and neonatal cord blood) to generate an HSC-hu mouse model. These CD34+ hu mice can be infected with HIV and upon ART initiation, viral load is suppressed [123]. HIV establishes reservoirs in various tissues, and macrophage reservoirs of HIV were detected in the spleen and bone marrow [124,125]. Limitations of this model include poor lymph node and spleen reconstitution in the mice, including limited development of the T cell, B cell and myeloid cells [126,127,128]. Reports of mucosal transmission has been shown to be dependent on the strain of mice and the origin of CD34+ HSC [129,130].

The development of humanized mice by Garcia’s group, called myeloid only mice (MoM) was instrumental in studying the HIV reservoir in macrophages in the absence of CD4+ T cells. These mice were generated in NOD.CB17-Prkdc^scid^/J mice (NOD/SCID) by transplanting with human CD34+ hematopoietic stem cells (HSC). These mice were reconstituted with B cells and myeloid cells and were completely devoid of T cells [131]. Using this model, authors further demonstrated that tissue macrophages are a reservoir of persistent HIV during suppressive ART [132].

The most robust humanized model of HIV is the bone marrow, liver, thymus (BLT) mice. Hu-BLT mice are generated by implanting human thymus and liver tissues under the kidney capsule of immunocompromised mice like NSG or C57BL/6 Rag2^−/−^γc^−/−^CD47^−/−^ (TKO) mice and injecting autologous CD34+ HSC simultaneously [128,133,134]. The main advantage of BLT mice is the presence of a human thymic environment and systemic human cell reconstitution, including the mucosal tissues, making it an excellent model for studying mucosal transmission of HIV [135,136,137,138,139,140]. The hu-BLT model is the most suitable for HIV persistence studies due to their ability to establish and maintain latent reservoirs after extended periods of ART [135,141,142]. In addition, they were also used to demonstrate the establishment of viral reservoirs in the CNS that persist during ART [143]. However, there are limitations of this model since they are susceptible to GvHD starting six months post reconstitution [144]. There are improved BLT mouse model incorporating human spleen called the bone marrow–thymus–liver–spleen (BLTS) mice that showed improved GvHD instance and better reconstitution of T cell, B cell and macrophages [144,145]. Regardless, hu-BLT/BLTS mice still require the use of fetal tissue which makes them harder to generate, in addition to other general limitations of humanized mice, such as their small size, limited lifespan and the variability in immune cell lineage differentiation during reconstitution [118].

Humanized mouse models are excellent in vivo models for initial HIV cure studies and can successfully mimic results observed in non-human primate models, as reported for the use of reverse latency agents [146].

### 7.3. Non-Human Primates

Non-human primates (NHPs) are the only other species that are naturally susceptible to HIV infection. There are different strains of Simian Immunodeficiency Virus (SIV) that causes different disease phenotypes depending on the species of NHPs, which have been reviewed recently [147].

In infectious SIV model, infection of NHP with SIV leads to disease progression and immunopathogenesis that bears close resemblance to HIV infection in humans. That includes CD4+ T cell depletion, viremia that can be controlled using ART and ATI resulting in viral rebound [147,148]. However, the use of cross species SIV is shown to be most effective in infection progressing to AIDS, i.e., infection with one strain of SIV will not cause severe disease in some NHP (natural host) but would cause AIDS in a different NHP (non-natural host) [149]. This reduced pathogenesis in natural host was predicted to be due to the lack of macrophage infection [150].

The level of macrophage infection in most NHP models are detectable but are highly elevated upon CD4+ or CD8+ T cell depletion [151]. Infection of macrophages is greatest in pathogenic SIV conditions in non-natural hosts [150]. Various macrophage-tropic SIV strains are discussed in the review by Moeser et al. [152]. Briefly, SIVmac239, SIVsm804E-CL757, SIV/17E-Fr and SIVsmE543-3 were shown to efficiently infect macaque macrophage in vivo, and SIVmac316 could not infect macaque macrophages. However, there are conflicting reports regarding SIVmac251 infecting macrophages without depletion of CD4+ or CD8+ T cells [151,152,153,154,155,156,157,158,159].

SIV-macaque models have been used to detect monocyte/macrophage reservoirs in ART suppressed hosts by using modified quantitative viral outgrowth assay (qVOA) [155,160,161]. These models were also used to study the establishment of myeloid HIV infection. There are reports showing macrophage reservoirs established due to phagocytosis of infected CD4+ T cells, as well as defining the role of circulating monocytes establishing SIV infection in the brain and neurodegenerative disorders [162]. Infecting macaques with neurotropic- and macrophage-tropic SIV strains show disease pathology of SIV encephalitis (SIVE), which is similar to HIV-associated neurocognitive disorders (HAND) in humans [163]. There are also various ongoing studies in various NHP models of HIV to study the establishment of myeloid/macrophage infection. Due to these similarities, the SIV-macaque model has been a valuable tool for developing ART, testing LRA, vaccine development and developing therapeutics in combating comorbidities associated with PLWH the quest to cure HIV.

## 8. The Challenge of HIV Reservoirs in the Central Nervous System

Similar to the notion discussed above, specialized macrophages that reside in the CNS constitute a heterogeneous mixed population of microglia, perivascular, meningeal and choroid plexus macrophages, in addition to monocytes with capacity to infiltrate into the brain under specific conditions [164]. For decades, the concept that microglia originated from circulating monocytes dominated despite initial consideration that they developed during embryogenesis [31,165]. It has recently become clear that microglia, similar to other tissue-resident macrophages, do not arise from blood monocytes but they are distinct functional populations [34,58,166,167]. These long-lived cells with self-renewal capacity and increased resistance to cytopathic effects constitute a formidable potential reservoir of persistent HIV infection. In addition, the long-believed “static status” of microglial cells has been demystified and its active role in homeostatic functions and tissue maintenance and integrity has been recently demonstrated [164,168]. Finally, the immune-privilege status of the brain was challenged after the discovery of a functional lymphatic system in the CNS that drains into peripheral lymph nodes [169,170].

All these advances in our knowledge of the brain-macrophage resident cells are just starting to influence our concept of how HIV infects, establishes productive infection, and maintains cellular reservoirs of HIV in the CNS. The recovery of HIV DNA from brain tissues isolated from autopsies of people living with HIV (PLWH) on suppressed ART, confirmed the presence of CNS reservoirs [171,172]. The question is then, how does HIV establish and maintain such reservoirs?

## 9. HIV Infection and Maintenance of Reservoirs in the Brain

We do not totally understand how HIV disseminates in the brain, although available literature suggests that multiple non-exclusive mechanisms may be involved [173]. However, recent clinical evidence suggests that HIV invasion of the CNS occurs as early as 8 days after estimated exposure, with similar observations reported in SIV-infected macaques, which appears to be associated with increased cellular infiltration [158,174,175,176]. Lack of an ideal model to investigate reservoirs in the CNS and strategies to target and reduce them have been hampered by the lack of physiologically relevant models to resemble the complex human brain and the associated pathologies.

HIV productive infection has been shown in microglial cells, perivascular macrophages and choroid plexus from PLWH [177]. Myeloid cells express low levels of the CD4 receptor and therefore are susceptible to infection. During early infection, main HIV strains are macrophage (M)-tropic with tropism for the CCR5 receptor, that is constitutively expressed in myeloid cells and can also be upregulated during monocyte differentiation [178]. A deeper characterization of brain resident macrophages could provide further insights into specific populations with enhanced susceptibility to HIV infection, similar to recent studies that found a population of gut-resident macrophages with increased CD4 expression in the intestine [179]. In addition, monocyte-derived macrophages that replenish the brain as a normal homeostatic process display increased susceptibility to HIV infection compared to circulating monocytes serving as the Trojan horse model of neuroinvasion [173,180]. Finally, multiple studies have shown that neurotropic viruses adapt and evolve increased capacity to infect macrophages and in fact, genetic compartmentalization is found within the virus from the cerebrospinal fluid (CSF) [177].

Our lab is working on a novel concept of viral dissemination involving platelets. In vitro studies demonstrated that platelets could interact with HIV by harboring HIV particles within their canalicular system or processed virions in endosomal compartments [181,182]. Studies in chronically infected PLWH showed that platelets could bind infectious HIV and infect macrophages [183,184]. More recently, we and others confirmed a potential critical role of platelets in HIV infectivity and latency. Real et al. confirmed that platelets from ART-suppressed PLWH harbor replication-competent HIV [185], and Simpson et al. showed that platelets can promote infectivity by forming complexes with monocytes and CD4+ T cells [186]. Platelet infection could result from active thrombopoiesis of HIV-infected megakaryocytes, as recently suggested or via spontaneous uptake of HIV virions by platelets in the circulation of viremic patients as subsets of platelets express DC-SIGN and CLEC-2 [181,185,187,188,189,190]. It is possible that both events contribute to HIV-positive platelets in viremic individuals, resulting in higher levels of platelet-bound virions pre-cART, and only a decrease, not eradication, following treatment. Nonetheless, these studies set up an interesting novel notion of viral dissemination to the CNS by HIV-infected platelet–monocyte complexes infiltration in the brain that is currently being investigated.

The introduction of highly active antiretroviral therapy (HAART) led to a remarkable reduction of HIV associated dementia (HAND) and control of HIV replication in the CNS [191,192]. However, in the post-HAART period, there has been a rise in milder forms of dementia possibly associated to toxicity related to the continuous use of ART [193,194,195,196,197]. Even in the context of plasma viral suppression, some PLWH experience CSF/plasma discordance, meaning that HIV RNA levels are higher in CSF than in plasma, raising concerns about drug penetrance in the brain and CSF viral escape [176,198,199,200,201,202]. Therefore, maintenance of viral reservoirs in the CNS may be through several mechanisms, including low-level ongoing viral replication or controlled migration of circulating infected cells to the brain as part of the normal immune surveillance [203,204,205,206]. In this regard, the HIV-infected CD14^+^CD16^+^ monocyte subpopulation preferentially migrates towards the CNS and contributes to viral persistence in the brain [207].

## 10. Reactivating Latency from Macrophage Reservoirs

ART maintains undetectable viremia in PLWH, although eradication of the virus is not possible due to the presence of latent reservoirs including cells from the monocyte/macrophage lineage. We have discussed above how macrophages are an important HIV reservoir as they have the capacity to harbor viruses and produce infectious virions. One of the most studied strategies for the cure of HIV is the so-called “shock and kill”, although mostly focus on targeting CD4+ T cell reservoirs [208]. For this purpose, latency reversal agents (LRAs) are used to reactivate latently infected cells, it had been proven that LRAs can efficiently reactivate latent CD4+ T cells [208]. However, evaluation of LRA function on macrophage populations is critical since the transcription factors that regulate HIV latency and maintenance in macrophages are distinct from those in CD4+ T cells [209,210]. In addition, inherent enhanced resistance to apoptosis and HIV-cytopathic effects [211], constitute critical differences compared to CD4+ T cell reservoirs that constitute additional challenges to achieve a complete cure.

A recent study by Hany et al. analyzed how different LRAs (bryostatin-1, JQ1 and romidepsin) affected human monocyte-derived macrophages [212]. They observed that the treatment with LRAs decreased macrophage susceptibility to HIV-1 infection. Furthermore, bryostatin-1 and romidepsin resulted in downregulation of CD4 and CCR5 receptors, respectively, that was accompanied by a reduction of R5 tropic virus infection. HIV-1 replication was mainly regulated by receptor modulation via bryostatin-1, while romidepsin effects rely on upregulation of SAMHD1 activity. However, these results conflict with a previous study by Campbell et al., showing that romidepsin did not affect CD4 or CCR5 expression in monocyte-derived macrophages [213].

Studies that evaluated the effect of Histone Deacetylase inhibitors (HDACi) in MDMs in vitro showed induction of an autophagy-dependent degradation of viral particles without altering the initial infection of macrophages [213]. This study showed that HDACi decrease HIV release from macrophages in a dose-dependent manner via degradation of intracellular HIV through the canonical autophagy pathway.

Another study using an in vitro primary human MDM model to study reactivation of HIV-1 transcription was used to evaluate latency modulating therapeutic agents [213]. FACS-sorted GFP–MDMs were cultured in the presence of T20 (enfuvirtide), with or without one of the LRAs: bryostatin, panobinostat or vorinostat, and the proportion of reactivated GFP+ cells was quantified. HIV reactivation was significantly increased in MDMs treated with bryostatin and vorinostat. Panobinostat did not significantly alter HIV reactivation in this model. However, LRA-induced reactivation, relative to spontaneous reactivation, differed between donors, suggesting variability in the susceptibility of the individual latent reservoirs to latency modulation [108]. These results suggest that similar to the lymphocyte reservoir, the heterogeneity of monocyte/macrophage subpopulations may have distinct susceptibility to reactivation.

Specific challenges associated with eradication of brain reservoirs have been recently reviewed and we discussed them briefly here [210,214]. Current LRAs that have progressed to clinical evaluation for an HIV cure have mostly been used against cancers and therefore, the information about neurotoxicity and BBB penetrance is available [210]. Overall, the administration of HDACi to PLWH was well tolerated and adverse events were mostly categorized as grade 1 or 2 with the exception of one grade 3 reported in a recent clinical trial [215,216,217,218]. One pilot study assessed the potential neurotoxicity of HDACi, panobinostat, showing that its administration did not induce adverse effects in the CNS [216]. However, the lack of detection of CSF HIV RNA, indicated that panobinostat may also have low BBB penetrance and therefore its utility for targeting CNS reservoirs may be limited. Increasing evidence from basic and preclinical studies demonstrates the need for careful examination of the neurotoxic effects of LRAs and their capacity to induce latency reversal both in lymphoid and myeloid cells [219,220,221,222,223].

The use of toll-like receptor (TLR) agonists as an additional class of LRA has been recently reviewed [224]. Due to their potential dual capacity to reactivate viral latency and induce immune responses, they constitute an attractive area for research. Their capacity to reverse latency has been demonstrated in in vitro studies, animal models and clinical trials. Their role as LRA was previously shown in HIV-infected monocytes/macrophages [224]. In latently infected monocytic cell lines, HIV replication was activated upon treatment with TLR-2, TLR-3 (Poly-I:C), TLR-9 (CpG) and TLR-7/8 (R-848) agonists [225,226,227,228,229]. LPS, a TLR-4 agonist, has also been shown to reactivate latent HIV macrophages from urethra of ART suppressed patients [82]. TLR-2 agonist molecules derived from *Mycobacterium Tuberculosis* have also been shown to reactivate HIV in latency models of microglia [106]. In the humanized mouse and Rhesus macaque models, treatment with (Poly-I:C) and TLR-7 agonist (GS-9620) has demonstrated a reduction in HIV reservoirs in numerous studies [230,231,232,233,234,235]. TLR-7 agonist (GS-9620) and TLR9 (MGN1703) have reached clinical testing in PLWH, although the specific effect on myeloid reservoirs has not yet been evaluated [236,237,238].

## 11. Concluding Remarks

As part of the “kill” portion of the shock and kill strategy, most studies of CD8+ cytotoxic T lymphocytes (CTLs) targeting HIV reservoirs have focused on CD4+ T cells, with few exceptions of targeting macrophage reservoirs. There are studies showing HIV/SIV-infected macrophages are resistant to CTL-mediated killing, even though they can effectively target CD4+ T cells [239,240,241]. This might be due to intrinsic difference in macrophages and CD4+ T cells. Macrophage killing was more dependent on the granzyme B and caspase-3 pathway, which were different from CD4+ T cell killing [239].

Other cytotoxic effector cells are also being investigated for killing HIV-infected macrophages. Natural killer cells were shown to have less effective cytolytic and ADCC response against HIV-infected macrophages compared to CD4+ T cells [242]. We have previously shown that another cytotoxic effector cell, gamma delta (γδ) T cells, have potentials for eliminating persistent HIV in lymphocyte reservoirs [243,244,245]. Our lab is currently exploring the ability of these cells to target non-T cell reservoirs while exploring the development of an immunotherapy for HIV cure [245].

In summary, a complete HIV cure will not happen without elimination of all reservoirs of persistent HIV. Therefore, it is critical to further characterize, investigate and analyze the impact of LRAs on myeloid populations, their susceptibility to reactivation and the source of viral rebound, and killing by effector cells.

## Figures and Tables

**Figure 1 pathogens-11-00611-f001:**
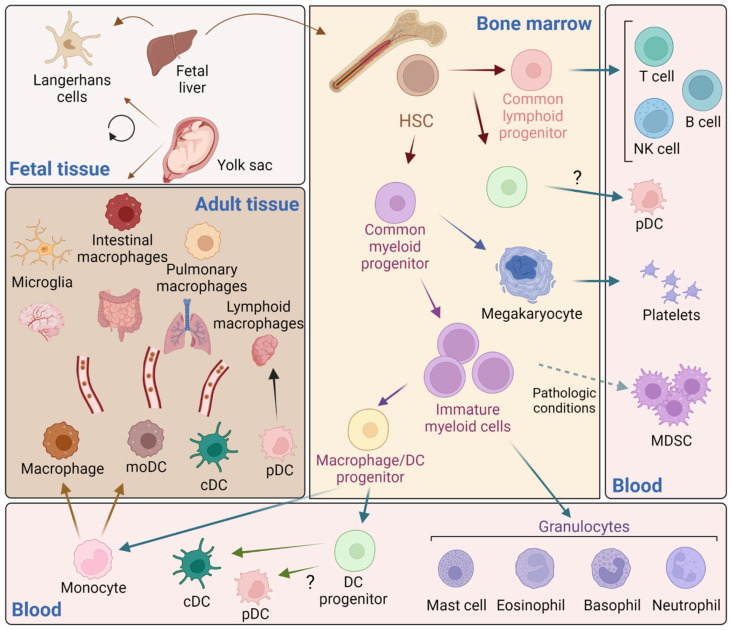
Schematic representation of white blood cells’ ontogeny. Tissue-resident macrophages with self-renewal capacity originate from the yolk sac (i.e., pulmonary, intestinal, brain (microglia), lymphoid, skin (Langerhans) during embryogenesis and populate their respective tissues. Langerhans cells also derive from the fetal liver, which produces hematopoietic stem cells (HCS) that colonize the bone marrow. HSC give rise to both common lymphoid and myeloid progenitors with capacity to produce megakaryocytes, precursors of platelets. Common myeloid progenitor cells also give rise to a heterogeneous population of immature myeloid cells that rapidly differentiate into granulocytes and a macrophage/dendritic cell (DC) precursor. Monocytes migrate to replenish the tissues and mature into macrophages and monocyte-derived DC (moDC) that as part of the normal homeostasis and surveillance. DC progenitors give rise to conventional DC (cDC) and plasmacytoid DC (pDC), although it is not clear whether pDC arise from an intermediate precursor prior to the generation of common lymphoid progenitors (recently reviewed in reference x). Under pathological conditions (e.g., infection, tumor, inflammation) immature myeloid cells produce myeloid-derived suppressor cells (MDSC) (Figure created with Biorender).

## Data Availability

Not applicable.

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
