# Peer review of "HIV Latency in Myeloid Cells: Challenges for a Cure"

_pathogens, 2022, doi:10.3390/pathogens11060611_

Round 1
Reviewer 1 Report
Comments and Suggestions for Authors
Manuscript Number: pathogens-1682558
Major comments:
The article entitled " HIV Latency in myeloid cells: challenges for a cure “has been reviewed. The manuscript started with a hypothesis that “HIV latently infected cells “are a reservoir and then concluded the correlation with eliminating HIV by using the “shock and kill” strategies to link with the initial hypothesis. The manuscript is well written and provides an extensive review of recent studies. I have two main concerns about this manuscript. First, the strategies to eliminate HIV latently infected cells may be not the aim of the study. The authors strongly confirmed that the authors described the role of the cytotoxic effector cell, gamma delta-T cells, which are related to killing HIV-infected macrophages in previous studies in conclusion. The conclusion of this study is a debate on 'shock and kill' strategies. There are other strategies to eliminate HIV latently infected cells.1,2,3 Secondary, the study enrolled 220 studies, the discussion was very extensive, it would be suggested to have a summary of components that included the types of cells and anatomic sites in HIV latently infected cells initially, induce apoptosis of HIV latently infected cells (pathogenesis), and then follow up certain studies such as clinical trials or medical studies, laboratory experiments, and medical procedures are performed in vivo or in vitro according to the “Cellular and Anatomical Reservoirs”, and finally the authors confirmed the novel conclusion.
- Yang H, Buisson S, Bossi G, Wallace Z, Hancock G, So C, Ashfield R, Vuidepot A, Mahon T, Molloy P, Oates J, Paston SJ, Aleksic M, Hassan NJ, Jakobsen BK, Dorrell L. Elimination of Latently HIV-infected Cells from Antiretroviral Therapy-suppressed Subjects by Engineered Immune-mobilizing T-cell Receptors. Mol Ther. 2016 Nov;24(11):1913-1925.
- Matsuda K, Kobayakawa T, Kariya R, Tsuchiya K, Ryu S, Tsuji K, Ishii T, Gatanaga H, Yoshimura K, Okada S, Hamada A, Mitsuya H, Tamamura H, Maeda K. A Therapeutic Strategy to Combat HIV-1 Latently Infected Cells With a Combination of Latency-Reversing Agents Containing DAG-Lactone PKC Activators. Front Microbiol. 2021 Mar 17;12:636276.
- Ahlenstiel, C. L., Symonds, G., Kent, S. & Kelleher, A. D. (2020). Block and lock HIV cure strategies to control the latent reservoir. Frontiers in cellular and infection microbiology, 10, 424.
Minor comments:
- In the Abstract, the description “However, other cell types, like circulating monocytes and tissue resident macrophages, can harbor integrated, replication competent HIV. ” is blurred. (P2 L61)
- In the Abstract, the description “shock and kill” approach” is (P2 L61), using the “shock and kill” strategy is more concise.
- In the Abstract, the description” However, other cell types, like circulating monocytes and tissue resident macrophages, can harbor integrated, replication competent HIV. To develop a cure for HIV, focus is needed not only on the T cell compartment, but also on these myeloid reservoirs of persistent HIV infection.” is blurred.
- In the Abstract, the description” HIV cure strategies and challenges associated to their identification and specific targeting by the “shock and kill” approach.” is blurred.
- Keywords: The HIV viral reservoirs would be more concise than「cellular reservoirs」. Are HIV persists or persistent HIV or HIV reservoir the same as HIV viral reservoirs? How many types of cells can be HIV viral reservoirs? The authors could be defined in the Introduction.
- Given that HIV viral reservoirs would have 'cellular and anatomy reservoirs', the readers could be confused. Also, we would suggest discussing this clearly separately in the study. [JAMA. 1998;280(1):67-71.]
- In the Introduction, the description 'However, ART does not eliminate HIV, and after depletion of CD4+ T cells during primary infection, some cells enter a resting state where HIV can remain persistently integrated into the host cell '
- In the Introduction, the description “Resting memory CD4+ T cells are the most widely studied cellular reservoir of persistent HIV and they are extremely stable over time.” is blurred.
- In the Introduction, the description “HIV persists in all subsets of CD4+ T cells including memory and naïve cell subpopulations although distinct transcriptional activity and integration sites can contribute to variable viral inducibility following ” is blurred.
- In the Introduction, the description “This article reviews main characteristics of human circulating and tissue resident myeloid cells, their role as reservoirs and potential issues for current strategies to cure HIV infection. ” is blurred.
- The description “Hematopoietic stem cells in the bone marrow produce two different progenitor cells that will give raise to the lymphoid and myeloid lineages (Figure 1). ” is blurred.
- As the first responders to infection, they are rapidly recruited to sites of infection and they are short-lived with a circulating half-life of 6-8 hours.
- The description “Monocytes constitute the circulating precursors of macrophages, which reside virtually in all tissues. ” is blurred.
- However, it has become clear now that both circulating monocytes recruited into tissues contribute to in situ homeostasis and many tissue-resident macrophage populations are located in those tissues and are homeostatically maintained
- This improved understanding of the immunobiology and function of the myeloid population has been fundamental in expanding investigations toward characterizing how the HIV viral reservoir is established, maintained and reactivated in specific monocytes and macrophage.
- The description “Further work has defined additional monocyte subpopulations with specific epigenetic and…” is blurred.
- The description “Classical monocytes express high levels of CCR2 and low levels of CX3CR1 and migrate in response to CCL2 chemokine gradients, while CD16+ monocytes express low CCR2 and high CXCR1 levels and migrate towards CXCL1 (fractalkine) gradients” is blurred.
- Expression patterns of these chemokine receptors among the three main monocyte subpopulations are linked to specific functions.
- The description “Tissue-resident macrophages have critical roles as first line of defense against intruding pathogens, maintaining homeostasis, and tissue integrity. ” is blurred.
- HSCs give rise to both common lymphoid and myeloid progenitors with the capacity to produce megakaryocytes, precursors of platelets.
- Although CD4+ T cells are the main target of HIV infection, macrophages express low levels of CD4 and their susceptibility to productive infection was described more than two decades ago.
- Macrophages likely play two opposing roles in the acute phase of HIV infection.
- Circulating monocytes in the blood have a shorter lifespan (~4 to 7days) and they differentiate into macrophage phenotype soon after their extravasation into the
- Their role in maintaining a persistent long-term HIV viral reservoir has therefore been controversial.
- Classical monocytes, expressing a low level of CD4 and CCR5 receptors, are less susceptible to HIV infection than intermediate or nonclassical monocytes.
- Therefore, the identification of the HIV viral reservoir within monocytes requires a deeper phenotypic characterization.
- The description “Early and late, but not integrated, HIV transcripts were detected in blood monocytes of a small population of ART- suppressed PLWH.” is blurred.
- The description “Even though the role of circulating monocytes as a reservoir of HIV is debatable, tissue resident macrophages have long known to be cellular reservoirs because of their presence in multiple tissues and their long lifespan. ” is blurred.
- The description “Gut macrophages were believed to be resistant to HIV infection and were thought to play a role in generating antibody responses” is blurred.
- ‘’..HIV RNA and proviral DNA were detected in alveolar macrophages in some but not other patients.’’
- The description “This suggests the latent alveolar macrophages are impairing pulmonary immunity making PLWH more susceptible to respiratory tract infections” is blurred.
- Although these cells exhibit a background level of spontaneous expression of GFP in culture, the latency of HC69 cells can be effectively reversed using TNFα and poly (I:C) TLR3 agonist.
- The description “「Human primary monocyte models」is blurred.
- The description “infected MDMs were cultured for up to 78 days post infection and observed that the FACS purified GFP- MDMs harbored proviral DNA, which could be reactivated by IL-4 and PMA to produce replication competent HIV” is blurred.
- The description “Human macrophages are also not incorporated in this model, which makes them a poor model for studying macrophage HIV reservoirs” is blurred.
- The description “The monocytic cell line (THP-1) was derived by culturing blood of a patient with acute monocytic leukemia by Tada’s group. ” is blurred.
- The description “Reports of mucosal transmission has been shown to be dependent on the strain of mice and the origin of CD34+ HSC” is blurred.
- These mice were reconstituted with B cells and myeloid cells and were completely devoid of T cells.
- The most robust humanized model of HIV is the bone marrow, liver, and thymus (BLT) mice.
- Similar to the notion discussed above, specialized macrophages that reside in the CNS constitute a heterogeneous…
- The description “This study set up an interesting novel notion of viral dissemination to the CNS by HIV-infected platelet-monocyte complexes infiltration in the brain that is currently being investigated. ” is blurred.
- Please confirm the wording ” towards” throughout the manuscript.
- However, in the post-HAART period there has been a rise of milder forms of dementia possibly associated to toxicity related to the continuous use of ART
- The description “However, evaluation of LRA function on macrophage populations is critical since the transcription factors that regulate HIV latency and maintenance in macrophages are distinct from those in CD4+ T cells” is blurred.
- The description “Another study using an in vitro primary human MDM model to study reactivation of HIV-1 transcription was used to evaluate latency modulating therapeutic agents” is blurred.
- The description “Specific challenges associated to eradication of brain reservoirs have been recently reviewed and we discussed them briefly herein” is blurred.
- The description “We have previously shown that another cytotoxic effector cell, gamma delta T cells, have potential to eliminate persistent HIV in lymphocyte reservoirs’’ would be the authors’ opinions.
- The description “Our lab is currently exploring the ability of these cells to target non-T cell reservoirs while exploring the development of an immunotherapy for HIV cure ’’ would be the authors’ opinions.

Author Response
Unfortunately, we do not understand what the first reviewer means with "blurred" and in general we are unsure what the reviewer is referring to.
Thank you,
Natalia Soriano-Sarabia

Reviewer 2 Report
The manuscript of Chitrakar et al “HIV Latency in myeloid cells: challenges for a cure” is a comprehensive review of potential role of myeloid cells, monocytes and macrophages, including microglia, as reservoirs of HIV infection. Traditionally, the resting memory CD4+ T cells are considered to be the major reservoir of persistent HIV and therefore a target for shock-and-kill therapies to eradicate HIV infection in the body. In many publications myeloid cells are not considered as reservoirs of latent HIV as there are no unambiguous data on the reestablishment of productive infection only from myeloid cells without contribution of CD4+ T cells when ART is stopped. On the other hand, macrophages and microglia infected with HIV and SIV may persist throughout ART, contributing to overall size of the latent reservoir and facilitate viral rebound after ART interruption that makes this review relevant and timely.
The review is well-written and logically organized. The authors elucidate major aspects of biology of myeloid cells, summarize available data on HIV infection in circulating monocytes and tissue-resident macrophages, consider in vitro and animal models for studying HIV and SIV infection in myeloid cells, reactivation, application of LRA for macrophages and particularly focus on HIV reservoirs in CNS. Overall, the manuscript is of interest and is supposed to be important not only for HIV biologists, but also for a wide range of virologists, immunologists and clinicians. The authors successfully combined and organized earlier published data along with very new findings. Without diminishing the undoubted scientific and educational value of this review, below I provide a few minor critical remarks.
- p. 3, second paragraph: “Classical monocytes migrate to sites of injury and differentiate into inflammatory macrophages, whereas non-classical monocytes exhibit immune surveillance functions along the blood vessels”. Since in previous paragraph the authors also identified subpopulation of intermediate monocytes, their function and further differentiation should be described here.
- p.5. It is not clear whether monocytes harbor HIV genomes and whether they can be productively infected. The authors indicate that “early and late, but not integrated, HIV transcripts were detected in blood monocytes of a small population of ART- suppressed PLWH”. The authors probably mean products of reverse transcription, not RNA transcripts. It is interesting observation that HIV provirus is not detected in monocytes. It would be interesting to discuss possible reasons for this phenomenon, in particular potential role of restriction factors like SAMHD1 at early stages of infection. Role of this restriction factor can also be considered when authors discuss HIV-1 infection in macrophages.
- p.6, second paragraph: “…there is significant evidence of the role of macrophages as reservoirs in ART-suppressed individuals that warrants further studies to characterize the role of macrophages in ATI interventions aimed at HIV eradication”. Based on numerous publications indicating that HIV-1 infection in macrophages is usually low-productive and the fact that macrophages, especially the resident ones, are considered to be quiescent cells until they are not activated by PAMPs, it would be logical to consider them as HIV reservoirs regardless of ART treatment. Especially in the CNS where the access of antiretrovirals is generally limited. Based on macrophage biology, such factors as pathogen- or damage-associated molecular patterns could be considered as major activators of this latent reservoir.
- p.10, first paragraph: “In vitro studies demonstrated that platelets can interact with HIV by harboring HIV particles within their canalicular system or processed virions in endosomal compartments. Studies in chronically infected PLWH showed that platelets could bind infectious HIV and infect macrophages”. Earlier publications (around 2000s) indicated that megakaryocytes as well as their progenitors can be non-productively infected with HIV-1. In this context, it would be interesting to speculate about possibility of these myeloid cells to be an additional potential viral reservoir, which can then produce HIV-infected platelets bringing the virus to macrophages.
Author Response
Re: Response to comments from reviewer 2
We would like to thank the reviewer for the comments. We have now addressed the questions and the changes are highlighted in yellow in this revised version of the manuscript. Our responses are below in blue.
Sincerely,
Natalia Soriano-Sarabia
The manuscript of Chitrakar et al “HIV Latency in myeloid cells: challenges for a cure” is a comprehensive review of potential role of myeloid cells, monocytes and macrophages, including microglia, as reservoirs of HIV infection. Traditionally, the resting memory CD4+ T cells are considered to be the major reservoir of persistent HIV and therefore a target for shock-and-kill therapies to eradicate HIV infection in the body. In many publications myeloid cells are not considered as reservoirs of latent HIV as there are no unambiguous data on the reestablishment of productive infection only from myeloid cells without contribution of CD4+ T cells when ART is stopped. On the other hand, macrophages and microglia infected with HIV and SIV may persist throughout ART, contributing to overall size of the latent reservoir and facilitate viral rebound after ART interruption that makes this review relevant and timely.
The review is well-written and logically organized. The authors elucidate major aspects of biology of myeloid cells, summarize available data on HIV infection in circulating monocytes and tissue-resident macrophages, consider in vitro and animal models for studying HIV and SIV infection in myeloid cells, reactivation, application of LRA for macrophages and particularly focus on HIV reservoirs in CNS. Overall, the manuscript is of interest and is supposed to be important not only for HIV biologists, but also for a wide range of virologists, immunologists and clinicians. The authors successfully combined and organized earlier published data along with very new findings. Without diminishing the undoubted scientific and educational value of this review, below I provide a few minor critical remarks.
- p. 3, second paragraph: “Classical monocytes migrate to sites of injury and differentiate into inflammatory macrophages, whereas non-classical monocytes exhibit immune surveillance functions along the blood vessels”. Since in previous paragraph the authors also identified subpopulation of intermediate monocytes, their function and further differentiation should be described here.
Thank you for this comment. We have now expanded on the effector functions of the subsets.
- p.5. It is not clear whether monocytes harbor HIV genomes and whether they can be productively infected. The authors indicate that “early and late, but not integrated, HIV transcripts were detected in blood monocytes of a small population of ART- suppressed PLWH”. The authors probably mean products of reverse transcription, not RNA transcripts. It is interesting observation that HIV provirus is not detected in monocytes. It would be interesting to discuss possible reasons for this phenomenon, in particular potential role of restriction factors like SAMHD1 at early stages of infection. Role of this restriction factor can also be considered when authors discuss HIV-1 infection in macrophages.
We agree with the reviewer and we have now included the effect of well-known restriction factors such as APOBEC and SAMHD1 to our manuscript.
- p.6, second paragraph: “…there is significant evidence of the role of macrophages as reservoirs in ART-suppressed individuals that warrants further studies to characterize the role of macrophages in ATI interventions aimed at HIV eradication”. Based on numerous publications indicating that HIV-1 infection in macrophages is usually low-productive and the fact that macrophages, especially the resident ones, are considered to be quiescent cells until they are not activated by PAMPs, it would be logical to consider them as HIV reservoirs regardless of ART treatment. Especially in the CNS where the access of antiretrovirals is generally limited. Based on macrophage biology, such factors as pathogen- or damage-associated molecular patterns could be considered as major activators of this latent reservoir.
This is also a good point to have into consideration as such, we have included the evidence on TLR agonists reactivating latency in the last section of the manuscript.
- p.10, first paragraph: “In vitro studies demonstrated that platelets can interact with HIV by harboring HIV particles within their canalicular system or processed virions in endosomal compartments. Studies in chronically infected PLWH showed that platelets could bind infectious HIV and infect macrophages”. Earlier publications (around 2000s) indicated that megakaryocytes as well as their progenitors can be non-productively infected with HIV-1. In this context, it would be interesting to speculate about possibility of these myeloid cells to be an additional potential viral reservoir, which can then produce HIV-infected platelets bringing the virus to macrophages.
We have elaborated further on this concept as well. Thank you.
